# Omics based approaches to decipher the leaf ionome and transcriptome changes in *Solanum lycopersicum* L. upon Tomato Brown Rugose Fruit Virus (ToBRFV) infection

**Aditi Padmakar Thakare, Maria Cristina Della Lucia, Chandana Mulagala, Giovanni Bertoldo, Massimo Cagnin, Piergiorgio Stevanato** [ID] *

Department of Agronomy, Food, Natural Resources, Animals and Environment (DAFNAE), University of Padova, Legnaro, Padua, Italy

* stevanato@unipd.it

**Data Availability Statement:** Data regarding the quantification of elements for leaf ionomics is given

## Abstract

The Tomato Brown Rugose Fruit Virus (ToBRFV) is a pathogen that mostly affects plants from the *Solanaceae* family. This virus severely affects the yield of tomato (*Solanum lycopersicum* L.) plants, thus creating an urgent need to research the basis of resistance to manage the disease. To understand the molecular basis of resistance, we employed omics-based approaches involving leaf ionomics and transcriptomics to help us decipher the interaction between elemental and nutritional composition and investigate its effect on the gene expression profile upon the ToBRFV infection in tomatoes. Ionomics was used to investigate the accumulation of trace elements in leaves, showcasing that the plants resistant to the virus had significantly higher concentrations of iron (p-value = 0.039) and nickel (p-value = 0.042) than the susceptible ones. By correlating these findings with transcriptomics, we identified some key genes involved in iron homeostasis and abscisic acid pathways, potentially responsible for conferring resistance against the pathogen. From the obtained list of differentially expressed genes, a panel of mutation profile was discovered with three important genes—Solyc02g068590.3.1 ($K^+$ transporter), Solyc01g111890.3.1 (LRR), and Solyc02g061770.4.1 (Chitinase) showing persistent missense mutations. We ascertain the role of these genes and establish their association with plant resistance using genotyping assays in various tomato lines. The targeted selection of these genetic determinants can further enhance plant breeding and crop yield management strategies.

## Introduction

Tomato (*Solanum lycopersicum* L.) is a crop of global significance with great economic and nutritional value. This crop is valued in terms of agricultural output for farmers, and contribution to international trade [1]. However, due to climate change plants may experience water, soil and temperature stress which in turn increase the chances of the plant to be more prone to

in supplementary table 1. The raw reads for transcriptome sequencing are submitted to ENA under project ID PRJEB72782.

**Funding:** This project has received funding from the European Union's Horizon 2020 research and innovation programme under the Marie Skłodowska Curie grant agreement no. 101034319 and from the European Union - NextGenerationEU.

**Competing interests:** The authors have declared that no competing interests exist.

pathogen attacks, thus affecting the yield and food security [2]. One such production limiting factor for plants are the viral attacks having multi-faceted impacts on plant health, produce quality, soil health etc. [3].

Over the years, tomato plants have been susceptible to many pathogen attacks, one of which is Tomato Brown Rugose Fruit Virus (ToBRFV), a plant virus that primarily affects tomato plants. It is a tobamovirus, a member of the Tobacco Mosaic Virus (TMV) family. The genome consists of a single-stranded, positive-sense RNA molecule having four open reading frames (ORFs) that encode distinct viral proteins [4, 5]. The longevity of the virus is also aided by its ability to remain for long periods of time in seeds, soil, and plant debris. ToBRFV symptoms include leaf curling, mosaic patterns on leaves and most prominently, brown wrinkled fruits [6]. The European and Mediterranean Plant Protection Organisation states that ToBRFV infection reduces fruit clusters during harvesting and affects 30 to 70 percent of tomato yield [7]. ToBRFV has been able to overcome the resistance genes Tm1, Tm2, and Tm2$^2$ that were previously introduced into the tomato lines to inhibit the viral infection by targeting the helicase domain and movement protein of TMV and ToMV from *S. habrochaites* and *S. peruvianum* [8], thus making the spread of disease more rampant and a virus of concern. In a recent study published by Wang et al. [9], high-throughput RNA-Seq was performed to profile the gene expression changes in tomato leaves upon ToBRFV infection which allowed the identification of differentially expressed genes that are potentially involved in the plant defence response. They reported that virus infection can activate the expression of PTI-related genes like MAPK, WRKY transcription factors and Ca2+ binding protein. Subsequently, this study also associated the up-regulation of RLK genes (like leucine-rich-repeat RLKs, lectin RLKs, proline-rich RLKs, etc.) and 1 down-regulated RLK gene (LRR RLK), involved in the tomato response to ToBRFV infection.

Since centuries, traditional breeding techniques have been used to enhance crop qualities and traits by mass selection and back-crossing. However, these methods are imprecise, have a limited gene pool, and restrict recombination [10, 11]. Nevertheless, with the advances of omics-based technologies, it is feasible to expedite the process of plant breeding and correlate the phenotype to genotype by understanding the underlying biological processes [12]. One such omics-based technique is ionomics, which represents the entire mineral and elemental composition of the desired plant system using the concepts of spectroscopy. Ionomics offers a high-throughput method to analyze multiple elements simultaneously, providing a systems-level understanding of nutrient dynamics. For instance, ionomics can reveal how plants respond to nutrient deficiencies or toxicities by altering their elemental profiles. Enhancing plant resistance to biotic and abiotic stresses, optimising fertiliser use, and boosting crop nutrition are all critical for sustainable agriculture and food security in the wake of climate change. These kinds of insights are very helpful in these areas [13, 14]. Quantitative Trait Loci (QTL) and genes influencing the mineral and nutritional content have been identified utilising leaf ionome analysis [15]; the data obtained from this process has been utilised to determine the genetic marker for subsequent breeding methods [13, 16]. In addition, RNA-sequencing can be used to find candidate genes that are differentially expressed under contrasting phenotypes. These genes can be used to develop molecular markers like Single Nucleotide Polymorphism (SNP) and can be linked to target traits for marker-assisted selection (MAS) in plant breeding [17].

Plant health and nutrition are pivotal to its immune system, influencing its ability to mount effective defence responses against pathogens and environmental stresses. Appropriate nutrition supports the integrity of physical barriers like the plant cell wall, which acts as the first line of defence against invading pathogens. This nutrient availability directly influences the production of hormones like salicylic acid (SA), jasmonic acid (JA), and ethylene (ET) that regulate plant immunity and orchestrate defence responses against pathogens [18]. Various

defence mechanisms have been developed by plants to combat against biotic and abiotic agents and important strategies involving iron homeostasis have been found to be essential as a regulator of the Reactive Oxygen Species (ROS) response, which serves as a secondary messenger for mitogen-activated protein kinases (MAPKs), cell wall strengthening etc. [19]. Therefore, studying the crosstalk of the plant immune system can help us identify key targets at the molecular level.

In this study, we employed various omics-based strategies, including ionomics, transcriptomics along with SNP identification, and genotyping to understand the molecular basis of resistance against the ToBRFV virus. The use of advanced omics-based technologies like Inductively Coupled Plasma Optical Emission Spectroscopy (ICP-OES) and Next Generation Sequencing (NGS) aided in establishing the link between leaf nutritional composition with differentially expressed genes and SNP profiles for wider region. Additionally, to gain further confidence over a large mutation profile, PCR techniques like rhAmp genotyping assay for SNP allelic discrimination were implemented to validate the key variants contributing to the genetic differences, thus allowing us to evaluate the plant phenotypic traits.

## Materials and methods

### Selection of plant material and sampling

The plant material used in this study belongs to a collection of the DAFNAE at the University of Padova and were grown according to the agricultural and local practices followed in the cultivation of breeding tomato lines and varieties in Sicily. San Marzano variety of tomato were grown in pots under uniform conditions and the leaves were infected with endemic Tomato Brown Rugose Fruit Virus (ToBRFV). Fresh leaf samples were collected from 6 control and 8 infected plants (See Fig 1A) and were immediately frozen in liquid nitrogen, followed by storage in -80°C for mRNA sequencing. Additionally, 12 lines of tomato with different types ("ciliegino", "datterino", and "tondo liscio"), from Sicily with varied genetic backgrounds were exposed to natural infection towards ToBRFV. The presence of virus was quantified using quantitative polymerase chain reaction (qPCR) for leaf apparatus. For ToBRFV quantification, CaTa28 primers and probes were used, recommended by the International Seed Federation (ISF). Samples with a Ct of 35 cycles or less were considered susceptible.

### Ionomic analysis

Six leaves from each of the twelve different lines of tomato were selected showcasing the resistant and susceptible traits (see Fig 1B). These leaf samples were dried overnight in the oven at a

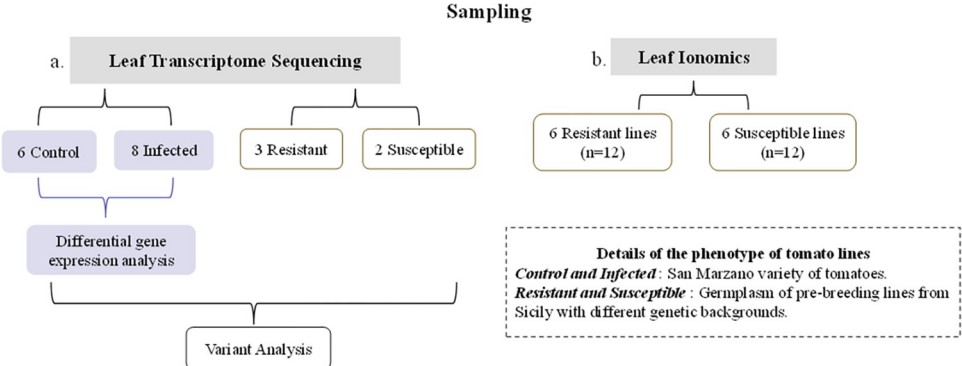

**Fig 1. Details of leaf samples used for transcriptomics, variant analysis and ionomics purposes.**

temperature of 95–100°C and weighed ~1.2 gm. Ionomic analysis was performed, and the element concentration was determined by inductively coupled plasma ICP-OES. Elements were quantified using certified multi-element standards. The content of Al, As, B, Ba, Be, Ca, Cd, Co, Cr, Cu, Fe, Hg, K, Li, Mg, Mn, Mo, Na, Ni, P, Pb, S, Sb, Se, Sn, Sr, Ti, Tl, V and Zn were evaluated through the ICP-OES optical system. Contents were considered in mg kg$^{-1}$ of dry matter. This method was adapted from our recent publication [20]. A custom R script was used to flag the data outliers, and the levels of Fe and Ni (p-value < = 0.05) were compared between the two sets of resistant and susceptible samples using the Wilcoxon test.

### RNA extraction and transcriptome sequencing

Leaf samples weighing approximately ~40 mg were lysed using the TissueLyser II, Qiagen and the total RNA was extracted using the Qiagen RNeasy Plant Mini Kit protocol. Quantitative and qualitative tests were performed on the extracted samples. The Qubit RNA High Sensitivity Assay was used to quantify the RNA concentration, and the quality of the RNA using the RNA Integrity Number equivalent (RINe) was determined by Agilent Tapestation RNA screentape analysis. Samples having a concentration of more than 10 ng/ul and a RINe value of 4 or higher were further selected for transcriptome sequencing. Paired-end Illumina sequencing with a read length of 150bp was performed to obtain FASTQ files for further analysis.

### Gene expression analysis of the transcriptome data

Quality check on the raw reads was performed using the tool FastQC v0.11.9 [21]. Low quality reads were removed using Trim Galore [22]; 15 and 10 bp were clipped from 5' and 3', respectively. The trimmed reads were aligned to the reference genome of *Solanum lycopersicum* SL4.0 using bowtie2 version [23] to obtain SAM and sorted BAM files, by following the standard samtools pipeline [24]. Raw read counts were calculated using bedtools multicov suite with ITAG4.0 as a reference GTF/GFF file and was used as an input for DESeq2 to obtain the differentially expressed genes with log$_2$ fold change of +/- 2 and p-value of less than 0.05. The gene expression levels were normalised by Transcripts Per Million (TPM).

### Variant analysis

Variant calling of the sorted BAM files was done using 'bcftools mpileup' and 'bcftools call' suite [25] against the SL4.0 reference to obtain the VCF files. The resulting variants were then filtered for parameters of QUAL> = 10, DP> = 3 and AF> = 0.5 for quality, depth across samples and allele-frequency, respectively. SnpEff [26] tool was used to annotate the variants and predict its functional effects. To find out the common variants, individual VCF files for healthy and diseased categories were combined using 'bcftools merge' command and only the variants present in more than 60% of samples per category were used for further analysis (see Fig 1A). Variants with 'HIGH' and 'MODERATE' effect spanning in the exonic regions were screened to get the mutation profile.

### Validation of SNPs using genotyping

Variants selected after bioinformatics analysis were genotyped using rhAmp assays (Integrated DNA Technologies, USA) on a wide range of lines including 14 lines with biological replicates. Sequences used to design the rhAmp assay primers for mono-allelic SNPs are mentioned in S1 Text. DNA from the samples was extracted and purified using BioSprint 96 workstation (QIAGEN, Germany) by the method described by Stevanato *et al*. 2015 [27]. Genotyping was

performed in a 5 μL reaction volume with 5–10 ng of DNA, 2.65 μL of rhAmp Genotyping Master Mix, 0.25 μL of rhAmp SNP assay mix, and 1 μL of nuclease-free water with thermocycler conditions given in Broccanello et al. 2018 [28]. Allelic calls were performed using Quant-Studio™ Design and Analysis software v1.4.3. Statistical significance was calculated by chi-square test. This method was adapted from our recent publication [29].

## Results

### Ionomic analysis

Ionomic analysis was conducted on twelve leaf samples of each resistant and susceptible phenotype to understand the role of trace elements involved in plant nutrition and immunity. From the PCA plot in Fig 2A, it can be seen that the PC1 (with 55.14% variance) with the elements Iron (Fe) and Nickel (Ni) has the most significant effect in terms of composition. Fig 2B shows that resistant and susceptible samples form a separate cluster (except for the outlier samples R6 and S6, which were not considered for further analysis). From the boxplots in Fig 2C and 2D, both Fe (p = 0.039) and Ni (p = 0.042) have higher content in the resistant group as compared to the susceptible. S1 Table shows the content of elements present in each leaf sample (including biological replicates).

### Identification of candidate genes from transcriptome data

To analyse the differentially expressed genes, mRNA sequencing was performed on the control and infected samples (see Fig 1, highlighted in purple). To create a read counts matrix, the paired-end reads from sequencing were aligned to the genome of *Solanum lycopersicum* SL4.0, and a vst (Variance Stabilising Transformation) function of DESeq2 was used to create the PCA plot. There is a clear demarcation between the control and infected samples, which form separate clusters in Fig 3.

Genes were screened into categories related to Pathogen Triggered Immunity (PTI), Effector Triggered Immunity (ETI), and other important pathways including iron absorption and cell wall formation in order to understand the essential pathways regulated. As a result of the pathogen invasion, the maximum number of both ETI and PTI pathway genes are upregulated in the infected samples, see heatmap (Fig 4). However, the control samples exhibit up-regulation of genes involved in basic Helix-Loop-Helix (bHLH) transcription factor (Solyc06g065040.4.1), chitinase (Solyc02g061770.4.1), and iron-sulphur assembly (Solyc09g009440.3.1). The rare genes that were found to be up-regulated in the control samples belonged to two groups: (i) the abscisic acid pathways (ABA); these included the abscisic acid and environment stress induced protein (Solyc02g084850.3.1), Calcium-dependent protein kinase (CDPK) (Solyc02g091500.1.1) and potassium ($K^+$) transporter (Solyc02g068590.3.1), which are down-stream of ABA and involved in ion efflux and stomata opening / closure; (ii) Leucine Rich Repeat (LRR) (Solyc02g071860.4.1, Solyc01g111890.3.1), which is known to initiate a plant defence response cascade against pathogens upon recognition [30].

### Variant analysis

Nine healthy samples (three resistant and six control) and ten diseased plants (two susceptible and eight infected) were used for variant analysis (see Fig 1A). The variants were filtered for QUAL> = 10, DP> = 3, and AF> = 0.5 using the standard VCF pipeline. In total, 320,064 and 311,595 variants were identified in healthy and diseased leaf samples respectively. To find out the mutations imparting probable resistance against the virus, unique variants were identified

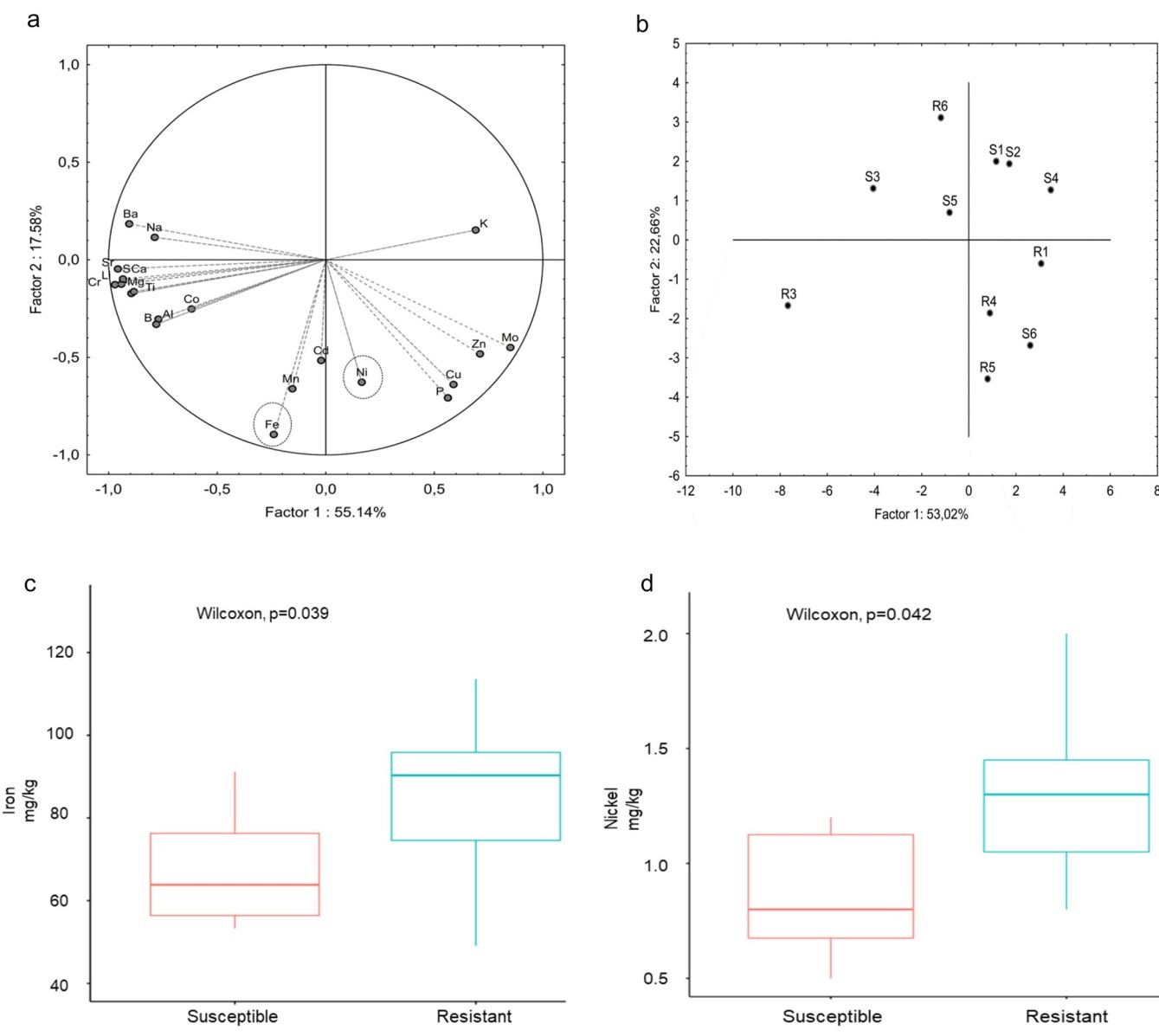

**Fig 2.** (a) PCA plot of ionomic samples from the leaf samples, highlighting iron and nickel, (b) Clustering of samples on grouping them by means, (c and d) Box plots the Fe and Ni concentrations respectively in susceptible and resistant samples.

for both groups given the condition that more than 60% of the samples harboured the same mutation. The two missense mutations seen in the healthy samples are related to potential genes implicated in plant immunity: (i) Solyc01g111890.3.1, which has been identified as an LRR gene, has an A to G missense mutation on chromosome 1, which results in an amino acid shift from serine to proline; (ii) Solyc02g068590.3.1 annotated as a $K^+$ transporter has an A to C alteration on chromosome 2 with a leucine to valine amino acid change. In addition, a significant insertion is present in the healthy leaf samples, one of which is located on Solyc06g065040.4.1 on chromosome 6 with a T to TC insertion in bHLH transcription factor. Significant mutations have also been found in three key genes of the diseased samples: (i) Solyc02g061770.4.1 known for its chitinase activity has an arginine to serine change due to A to C missense mutation; (ii) Solyc09g009440.3.1, a critical gene implicated in the Fe-S

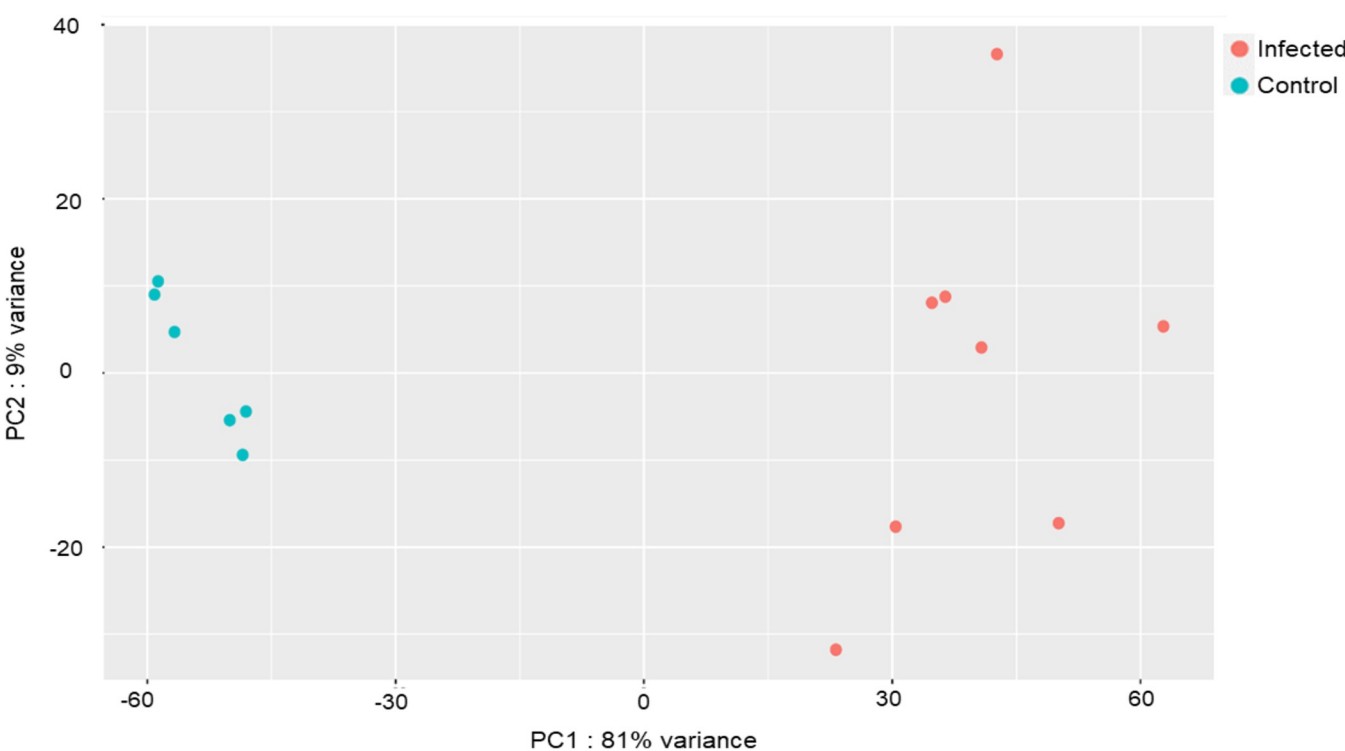

**Fig 3. PCA clustering showing that the artificially infected samples separate well from the controlled samples.**

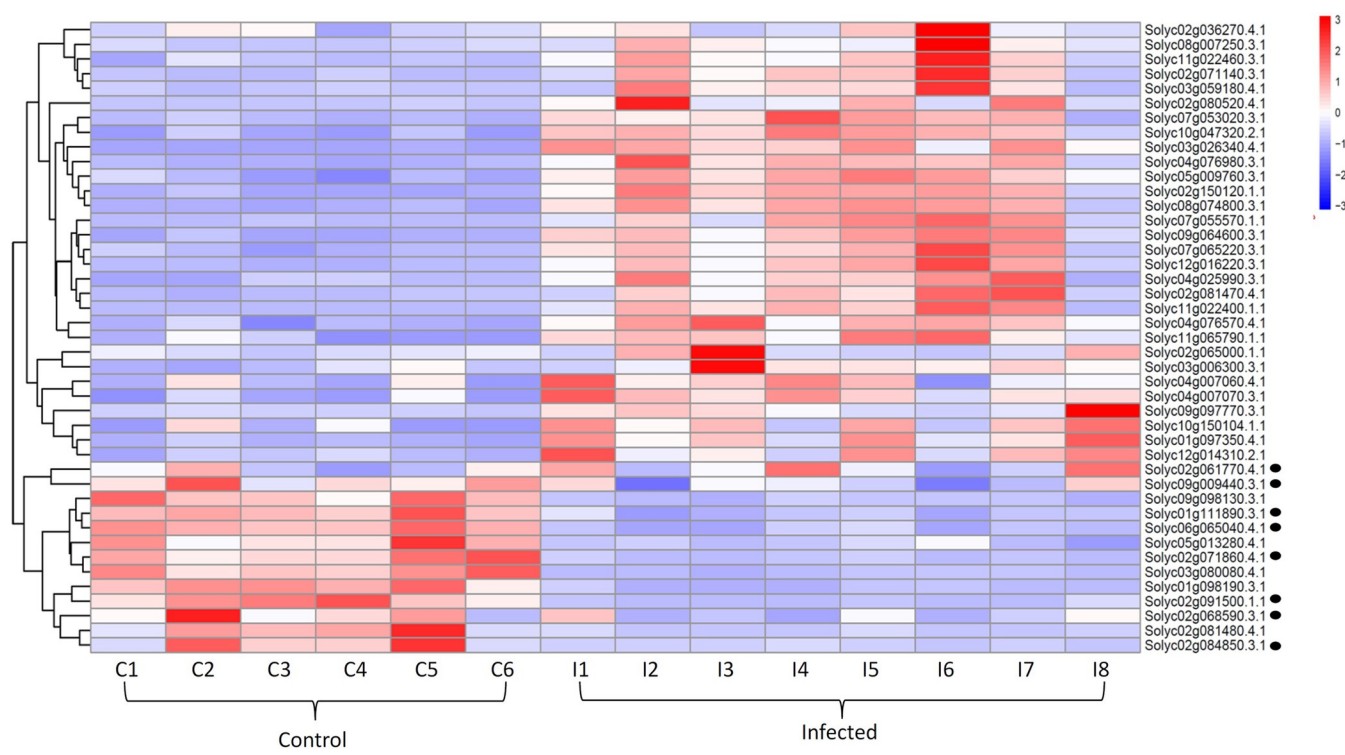

**Fig 4. Heatmap of differentially expressed genes between the control and infected samples based on TPM values.**

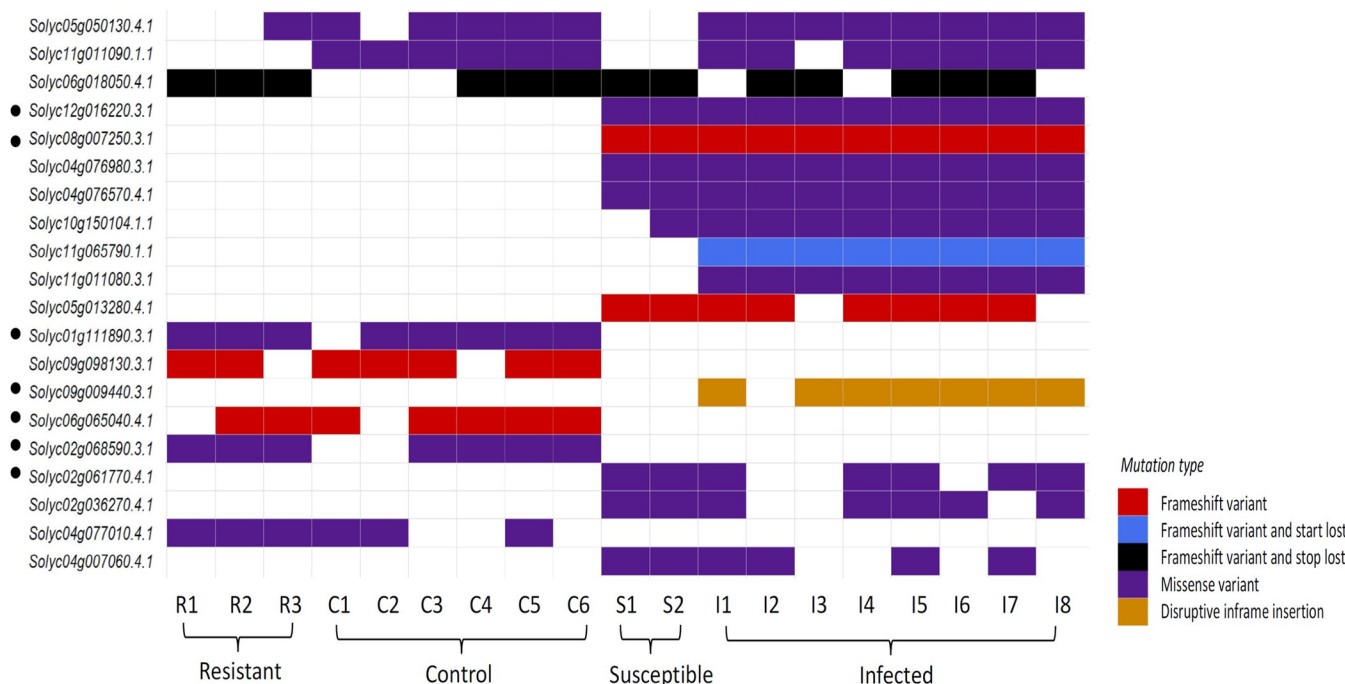

**Fig 5. The waterfall plot represents the mutation type in a list of genes for every individual sample.** The bars with no colour indicate that the sample has the same allele as the reference genome.

assembly, contains a large insertion on chromosome 9 that changes C to CGATGAGAAG, and (iii) the genes Solyc12g016220.3.1 and Solyc08g007250.3.1 that confer disease resistance have missense, frameshift, and insertion variants. Details of these SNPs along with their genome positions are described in S2 Table. A waterfall plot depicting the type of mutation per sample per category for a list of candidate genes is shown in Fig 5.

## SNP validation using rhAmp genotyping assays

From the above listed SNPs, three candidate genes with mono-allelic mutations were selected for validation using rhAmp genotyping assays from a large number of resistant and susceptible lines, respectively (see Fig 6). The allele frequency and chi-square test results (see Table 1) confirm the findings from the variant analysis that the genes Solyc02g061770.4.1- Chitinase, Solyc01g111890.3.1- LRR and Solyc02g068590.3.1- $K^+$ transporter harbouring homozygous alleles T/T, C/C and C/C were associated with disease resistance against ToBRFV exposed plants. S3 Table contains the details of the rhAmp assays including sample details, plant phenotype and genotype calls.

## Discussion

Plant pathogens wreak havoc affecting the yield and nutritional profile of the crops, and with the increasing rapid spread of the ToBRFV virus in recent years, it is necessary to understand the plant pathology and mechanisms to discover reliable genetic markers. To study this, we employed various omics-based approaches like ionomics, transcriptomics, SNP profiling to identify robust genetic markers against the ToBRFV infection. These findings showcase an example on how factors responsible for plant nutrition and genetics work in a conjoint manner to confer resistance against invasive pathogens.

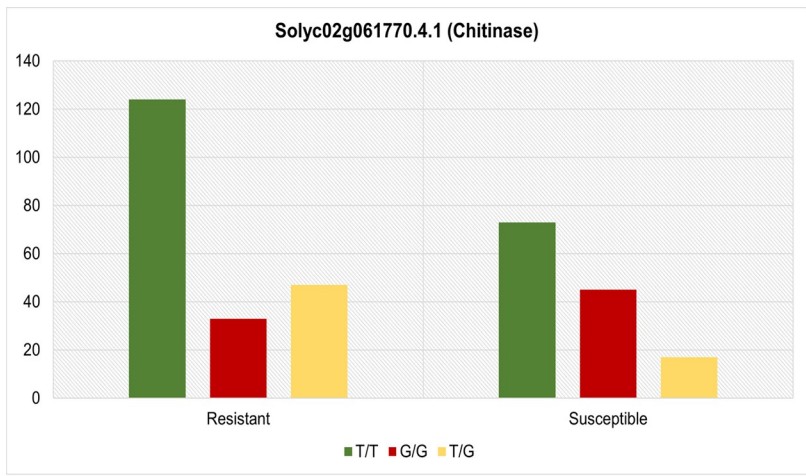

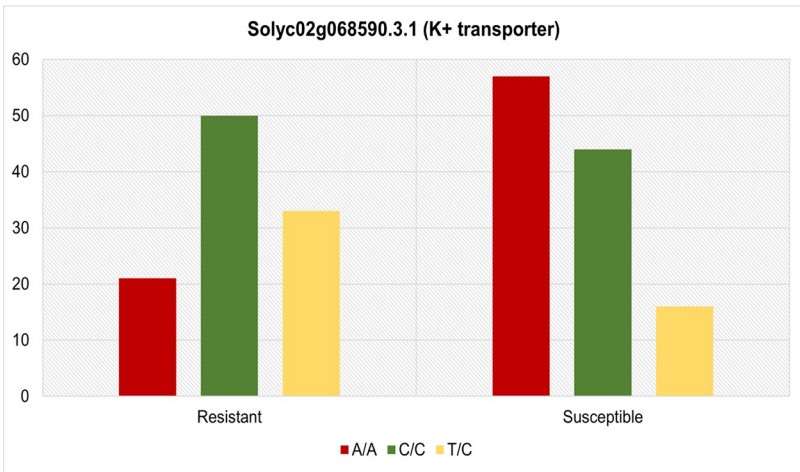

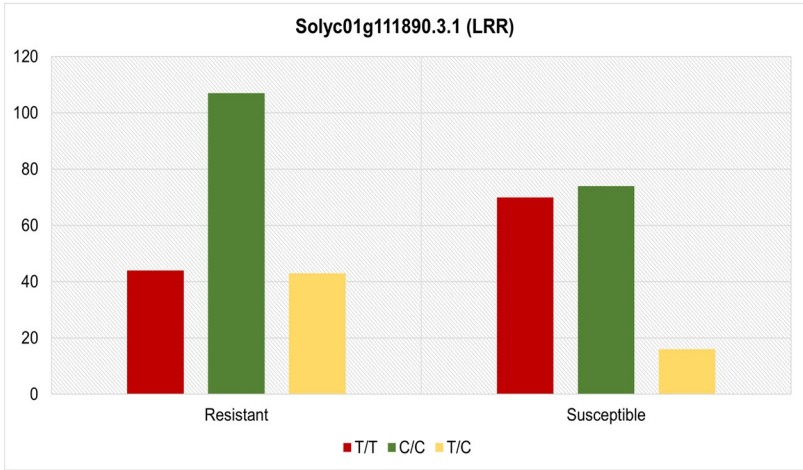

**Fig 6. Genotype frequencies of validated SNP targets using rhAmp assay in plants with resistant and susceptible lines.**

**Table 1. Allele frequency distribution of the validated SNPs through rhAmp genotyping assay and the calculated chi-square test significance.**

| Solyc02g061770.4.1 (Chitinase) | Samples | Phenotype | T/T | G/G | Total allele(2n) | Chi-square | p-val | Significant at p-val < 0.01 |
|---|---|---|---|---|---|---|---|---|
| | 204 | Resistant | 295 | 113 | 408 | 10.55 | 0.0011 | Yes |
| | 135 | Susceptible | 163 | 107 | 270 | | | |
| Solyc01g111890.3.1 (LRR) | Samples | Phenotype | T/T | C/C | Total allele(2n) | Chi-square | p-val | Significant at p-val < 0.01 |
| | 194 | Resistant | 131 | 257 | 388 | 16.34 | 0.00005 | Yes |
| | 160 | Susceptible | 156 | 164 | 320 | | | |
| Solyc02g068590.3.1 (K⁺ transporter) | Samples | Phenotype | A/A | C/C | Total allele(2n) | Chi-square | p-val | Significant at p-val < 0.01 |
| | 104 | Resistant | 75 | 133 | 208 | 16.83 | 0.00004 | Yes |
| | 117 | Susceptible | 130 | 104 | 234 | | | |

From the ionomics analysis, it is evident that the resistant variety had higher amounts of iron than the susceptible ones. Additionally, the transcriptomic analysis indicates that both the bHLH TF (Solyc06g065040.4.1) and Fe-S assembly proteins (Solyc09g009440.3.1) are upregulated with more copies of mRNA in the control group in contrast to the infected samples. The role of iron in plant nutrition and immunity influences the activity of receptors involved in pattern recognition receptors (PRRs) triggering the immune response. It is also interesting to note that iron is known to promote reactive oxygen species (ROS) and a hypersensitive response (HR) in plants upon infections [19]. Besides, the bHLH transcription factor plays a role in the uptake of iron, and Fe-S assembly is known to confer resistance by accumulating the metabolites relevant to defence mechanisms [31, 32]. These findings lead us to consider that the mutations in the bHLH and Fe-S assembly proteins (see S2 Table) obtained from variant analysis may have undergone structural modifications in order to increase iron absorption and confer resistance.

In combination with iron, a similar trend was observed in the ionome of tomato leaves of the resistant variety, wherein the absorption of nickel was comparatively higher than the susceptible plants. It is well known that plants produce methylglyoxal (MG) in response to stress, and nickel activation is necessary to detoxify the effects of it [33]. Furthermore, nickel is also involved in the synthesis of phytoalexins, which directly correlates with the production of lignin to strengthen the cell wall [34]. We suspect that higher Ni concentrations in the resistant plants helped detoxify the effect of MG to maintain a healthy metabolism, and the thicker cell wall acts as a primary defence response against the viral penetration. These findings demonstrate Ni as an essential element involved in defending against toxic compounds at the molecular and physiological levels.

As mentioned earlier, transcriptomics offers a quantitative insight on changes in the gene expression profile. This not only uncovers the novel genes implicated in pathways, but also provides candidate genes for marker-assisted selection [35]. According to our differential gene expression analysis, the Helix-Loop-Helix TF (bHLH) (Solyc06g065040.4.1), iron-sulphur assembly (Solyc09g009440.3.1), and Chitinase (Solyc02g061770.4.1) are up-regulated in the control samples, and are known to activate the hypersensitive response in plants to carry out the lysis or degradation of the pathogen [36]. Likewise, the LRR genes (Solyc02g071860.4.1, Solyc01g111890.3.1) are also elevated in control samples and are crucial in mediating effector- and PAMP-triggered immunity. More importantly, we hypothesise that, in agreement with the study conducted by Bharath *et al.* [37], the abundance of the ABA gene (Solyc02g084850.3.1) in the control samples may have enabled the outflux of potassium through the K⁺ transporter (Solyc02g068590.3.1) in combination with the upregulated CDPK gene (Solyc02g091500.1.1). This latter gene suppresses the K+ influx and keeps the stomata closed to prevent pathogen entry.

We identified SNPs in functional genes with the use of mRNA sequencing, which can be used to call genotypes [38]. To establish this, we used the rhAmp genotyping assays to precisely detect and distinguish the SNPs according to the phenotype. From a panel of mutation profile (see S2 Table), we narrowed down to three key genes i.e Solyc02g068590.3.1 (K+ transporter), Solyc01g111890.3.1 (LRR) and Solyc02g061770.4. (chitinase) which were further validated. In addition to being differentially regulated, these genes also harbour mono-allelic missense mutations, which can serve as a valuable genetic determinant and can further be extrapolated to create resilient tomato lines against ToBRFV.

## Conclusion

To our knowledge, this is the first instance where the genes pertaining to plant immunity against ToBRFV infection have been reported. The markers discovered for SNP identification have a strong correlation with differential gene regulation, thus validating its function in plant defence. We believe that these markers can serve as useful and functional targets in future to breed and establish disease resistance tomato cultivars.

## Supporting information

**S1 Fig. Allele discrimination plots for lines L1 and L13.**
(TIF)

**S2 Fig. Allele discrimination plots for line L4.**
(TIF)

**S1 Text. Sequences of the genes which were used to design rhAmp assay primer.**
(DOCX)

**S1 Table. Represents the leaf ionomics results.**
(XLSX)

**S2 Table. Shows candidate SNP positions from variant analysis.**
(XLSX)

**S3 Table. Represents the rhAmp genotyping results.**
(DOCX)

**S4 Table. Has all the qPCR Ct values for the leaf samples used in this study.**
(XLSX)

**S5 Table. Has the TPM values for the samples used for identification of candidate genes for differential gene expression patterns.**
(XLSX)

## Acknowledgments

The authors thank BMR Genomics SRL and Novogene UK Ltd. for generating Illumina transcriptome reads.

## Author Contributions

**Conceptualization:** Piergiorgio Stevanato.

**Formal analysis:** Aditi Padmakar Thakare.

**Funding acquisition:** Piergiorgio Stevanato.

**Methodology:** Aditi Padmakar Thakare, Maria Cristina Della Lucia, Chandana Mulagala, Massimo Cagnin.

**Software:** Aditi Padmakar Thakare.

**Supervision:** Piergiorgio Stevanato.

**Validation:** Aditi Padmakar Thakare, Chandana Mulagala.

**Writing – original draft:** Aditi Padmakar Thakare.

**Writing – review & editing:** Maria Cristina Della Lucia, Giovanni Bertoldo, Piergiorgio Stevanato.

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
