## [Decision Letter · Decision Letter 0]

7 Aug 2024

PONE-D-24-28075Omics based approaches to decipher the leaf ionome and transcriptome changes in Solanum lycopersicum L. crop upon Tomato Brown Rugose Fruit Virus (ToBRFV) infectionPLOS ONE

Dear Dr. Stevanato,

Thank you for submitting your manuscript to PLOS ONE. After careful consideration, we feel that it has merit but does not fully meet PLOS ONE’s publication criteria as it currently stands. Therefore, we invite you to submit a revised version of the manuscript that addresses the points raised during the review process. Please submit your revised manuscript by Sep 21 2024 11:59PM. If you will need more time than this to complete your revisions, please reply to this message or contact the journal office at plosone@plos.org. Please include the following items when submitting your revised manuscript:A rebuttal letter that responds to each point raised by the academic editor and reviewer(s). You should upload this letter as a separate file labeled 'Response to Reviewers'.A marked-up copy of your manuscript that highlights changes made to the original version. You should upload this as a separate file labeled 'Revised Manuscript with Track Changes'.An unmarked version of your revised paper without tracked changes. You should upload this as a separate file labeled 'Manuscript'.

We look forward to receiving your revised manuscript.

Kind regards,

Mojtaba Kordrostami, Ph.D.

Academic Editor

PLOS ONE

Journal Requirements:

   "NO"

   "This project has received funding from the European Union’s Horizon 2020 research and innovation programme under the Marie Skłodowska Curie grant agreement no. 101034319 and from the European Union - NextGenerationEU."

Reviewers' comments:

Reviewer's Responses to Questions

**Comments to the Author**

1. Is the manuscript technically sound, and do the data support the conclusions?

Reviewer #1: No

Reviewer #2: Yes

2. Has the statistical analysis been performed appropriately and rigorously? 

Reviewer #1: N/A

Reviewer #2: Yes

3. Have the authors made all data underlying the findings in their manuscript fully available?

Reviewer #1: No

Reviewer #2: Yes

4. Is the manuscript presented in an intelligible fashion and written in standard English?

Reviewer #1: Yes

Reviewer #2: Yes

5. Review Comments to the Author

Reviewer #1: The manuscript "Omics based approaches to decipher the leaf ionome and transcriptome changes in

Solanum lycopersicum L. crop upon Tomato Brown Rugose Fruit Virus (ToBRFV)" where author employed omics-based approaches like leaf ionomics and transcriptomics to decipher the interaction between elemental and nutritional composition and investigated gene expression profile upon the ToBRFV infection in tomatoes.

The experiment for transcriptomic data in the manuscript is not well planned. The experimental plant materials are not described. What are the resistant and susceptible cultivars has been used in the study.

The introduction is not nicely written and many recent and relevant references for ToBRFV resistance are missing.

It is not clear that what plant material has been used for transcriptomic experiment, resistant and susceptible lines or control and infected plants.

Line'129': reference is missing for the involvement of these LRR (Solyc02g071860.4.1, Solyc01g111890.3.1) genes in defense response against pathogen recognition.

Supplementary data for differential expressed genes with TPM value and annotation is missing.

Line '141-151': Not enough data has been provided to support the claim.

The author has not provided qPCR data for virus detection.

Reviewer #2: The manuscript describes a technically sound piece of scientific research with data that supports the conclusions. Experiments have been conducted rigorously, with appropriate controls, replication, and sample sizes. The conclusions has been drawn appropriately based on the data presented.

However, in title technical name of crop should be avoided. Just write tomato inplace of Solanum lycopersicum L.

Normally Materials and Methods should appear after Introduction, but here you are mentioning it after Conclusion. Why?

Add comma(,) in line 267 after 3'.

6. PLOS authors have the option to publish the peer review history of their article (what does this mean?). If published, this will include your full peer review and any attached files.

Reviewer #1: No

Reviewer #2: **Yes: **Prof. Pritam Kalia

---

## [Author Response · Author response to Decision Letter 0]

18 Sep 2024

Thank you for offering us an opportunity to respond to reviewer’s comments. We tried to answer every reviewer's comments independently and to the best of our ability we have addressed all the concerns and have made all necessary changes to the manuscript, which are tracked. We are thankful to the reviewers for such a thorough review, which has been both educational and improved the quality of the manuscript.

Sincerely,

Piergiorgio Stevanato

Journal guidelines and comments that are now revised : 

● Manuscript meets the PLOS ONE’s style requirements. 

● The funding section has been removed from the manuscript, and has been mentioned only in the online submission form.

● Role of the funders have been disclosed in the online submission form. 

● Captions for the ‘Supporting Information’ has been added at the end of the manuscript.

Reviewer #1: 

Q1 :The manuscript "Omics based approaches to decipher the leaf ionome and transcriptome changes in Solanum lycopersicum L. crop upon Tomato Brown Rugose Fruit Virus (ToBRFV)" where author employed omics-based approaches like leaf ionomics and transcriptomics to decipher the interaction between elemental and nutritional composition and investigated gene expression profile upon the ToBRFV infection in tomatoes. The experiment for transcriptomic data in the manuscript is not well planned. The experimental plant materials are not described. What are the resistant and susceptible cultivars has been used in the study ?

It is not clear that what plant material has been used for transcriptomic experiment, resistant and susceptible lines or control and infected plants.

A1 :We thank the reviewer for this thorough revision and suggestion. 

The plant material used in this study belongs to a collection of the DAFNAE at the University of Padova, and were grown according to the agricultural and cultivation practice of tomato breeding in Sicily. We used the San Marzano variety of tomatoes that were infected with endemic ToBRFV to monitor changes. Subsequently, 6 control & 8 infected phenotypes of the San Marzano plants were subjected to mRNA sequencing to study differential gene expression patterns.

We also evaluated germplasm of 12 lines of tomato with varied genetic backgrounds and different types like (“ciliegino”, “datterino”, and “tondo liscio”) which were grown and exposed to natural infection towards ToBRFV. The resistant & susceptible phenotypic status was assigned to the individual plant material based on the Ct values after performing qPCR. Five of these plants (3 resistant & 2 susceptible) were then subjected to mRNA sequencing, and analysed for variant analysis along with San Marzano samples. 

The reason to not include the above-mentioned five plants in the differential gene expression analysis, but only in variant analysis is due to the difference in library preparation, sequencing depth and smaller sample size. 

We have added a note regarding the plant materials in the updated Figure 6, and have also rephrased the sentences in the ‘Selection of plant material and sampling’ section of Materials and Methods. 

Q2 : The introduction is not nicely written and many recent and relevant references for ToBRFV resistance are missing.

A2 : We have now added more parts to the introduction section. The added sections emphasize on the recent findings done by (i) Wang et.al (https://www.mdpi.com/1422-0067/25/7/4012) on tomato plant response to ToBRFV infection; (ii) the role of ionomics in understanding plant nutrition dynamics, and (iii) pathways involved in plant immunity and defence responses. 

All these additions are highlighted as tracked changes. 

Q3 : Line'129': reference is missing for the involvement of these LRR (Solyc02g071860.4.1, Solyc01g111890.3.1) genes in defence response against pathogen recognition.

A3 : We have added the reference of Padmanabhan et. al, (https://onlinelibrary.wiley.com/doi/10.1111/j.1462-5822.2008.01260.x). The line numbers will be changed in both the tracked (line no. 147) & clean untracked version (line no. 215) of the manuscript.

Q4 : Supplementary data for differential expressed genes with TPM value and annotation is missing.

A4 : The data for the TPM values is now provided in Supplementary Table 5.

Q5 : Line '141-151': Not enough data has been provided to support the claim.

A5 : These mutations were obtained using variant analysis performed by 'bcftools' suite & annotated using 'SnpEff'. The details of the SNP position, amino acid change, mutation type etc. are provided in the supplementary table 2. 

Q6 : The author has not provided qPCR data for virus detection.

A6 : The Ct values for the qPCR performed on all the leaf samples is now provided in supplementary table 4.

Reviewer #2: 

The manuscript describes a technically sound piece of scientific research with data that supports the conclusions. Experiments have been conducted rigorously, with appropriate controls, replication, and sample sizes. The conclusions has been drawn appropriately based on the data presented.

Q7 : However, the technical name of the crop should be avoided. Just write tomato in place of Solanum lycopersicum L.

A7 : We thank the reviewer for this suggestion. We have now removed the word “crop” from the title. To avoid using the word "tomato" repeatedly next to each other, all the authors would like to retain the name 'Solanum lycopersicum L.' in the title. 

Q8 : Normally Materials and Methods should appear after Introduction, but here you are mentioning it after Conclusion. Why?

A8 : We apologise for misunderstanding the structure. We have now adhered to the PLOS guidelines for the manuscript organisation in the clean, untracked version of the manuscript titled 'Manuscript_Untracked17Sept2024'.

Q9 : Add comma(,) in line 267 after 3'.

A9 : We have added the “,” after 3’. The line number will be changed in the revised version (line no. 299) and in the clean untracked version (line no. 147) of the manuscripts.

---

## [Decision Letter · Decision Letter 1]

23 Oct 2024

Omics based approaches to decipher the leaf ionome and transcriptome changes in Solanum lycopersicum L. upon Tomato Brown Rugose Fruit Virus (ToBRFV) infection

PONE-D-24-28075R1

Dear Dr. Stevanato,

We’re pleased to inform you that your manuscript has been judged scientifically suitable for publication and will be formally accepted for publication once it meets all outstanding technical requirements.

Kind regards,

Mojtaba Kordrostami, Ph.D.

Academic Editor

PLOS ONE

Additional Editor Comments (optional):

Reviewers' comments:

Reviewer's Responses to Questions

**Comments to the Author**

1. If the authors have adequately addressed your comments raised in a previous round of review and you feel that this manuscript is now acceptable for publication, you may indicate that here to bypass the “Comments to the Author” section, enter your conflict of interest statement in the “Confidential to Editor” section, and submit your "Accept" recommendation.

Reviewer #3: (No Response)

2. Is the manuscript technically sound, and do the data support the conclusions?

Reviewer #3: (No Response)

3. Has the statistical analysis been performed appropriately and rigorously? 

Reviewer #3: (No Response)

4. Have the authors made all data underlying the findings in their manuscript fully available?

Reviewer #3: (No Response)

5. Is the manuscript presented in an intelligible fashion and written in standard English?

Reviewer #3: (No Response)

6. Review Comments to the Author

Reviewer #3: The authors have implemented the desired changes to a satisfactory degree, and the responses they have received are deemed appropriate.

7. PLOS authors have the option to publish the peer review history of their article (what does this mean?). If published, this will include your full peer review and any attached files.

Reviewer #3: No

---

## [Editor Report · Acceptance letter]

29 Oct 2024

PONE-D-24-28075R1 

PLOS ONE

Dear Dr. Stevanato, 

I'm pleased to inform you that your manuscript has been deemed suitable for publication in PLOS ONE. Congratulations! Your manuscript is now being handed over to our production team.

Kind regards, 

on behalf of

Dr. Mojtaba Kordrostami 

Academic Editor

PLOS ONE